# Impact Assessment of Pituitary Adenylate Cyclase Activating Polypeptide (PACAP) and Hemostatic Sponge on Vascular Anastomosis Regeneration in Rats

**DOI:** 10.3390/ijms242316695

**Published:** 2023-11-24

**Authors:** Laszlo Adam Fazekas, Balazs Szabo, Vince Szegeczki, Csaba Filler, Adam Varga, Zoltan Attila Godo, Gabor Toth, Dora Reglodi, Tamas Juhasz, Norbert Nemeth

**Affiliations:** 1Department of Operative Techniques and Surgical Research, Faculty of Medicine, University of Debrecen, Moricz Zsigmond ut 22, H-4032 Debrecen, Hungary; fazekas.laszlo@med.unideb.hu (L.A.F.); balazs.szabo@med.unideb.hu (B.S.); varga.adam@med.unideb.hu (A.V.); 2Department of Anatomy, Histology and Embryology, Faculty of Medicine, University of Debrecen, Nagyerdei krt. 98, H-4032 Debrecen, Hungary; szegeczki.vince@anat.med.unideb.hu (V.S.); filler.csaba@med.unideb.hu (C.F.); juhaszt@anat.med.unideb.hu (T.J.); 3Department of Information Technology, Faculty of Informatics, University of Debrecen, Kassai ut 26, H-4028 Debrecen, Hungary; godo.zoltan@inf.unideb.hu; 4Department of Medical Chemistry, Albert Szent-Györgyi Medical School, University of Szeged, Dom ter 8, H-6720 Szeged, Hungary; toth.gabor@med.u-szeged.hu; 5HUN-REN-PTE PACAP Research Group, Department of Anatomy, Medical School, University of Pecs, Szigeti ut 12, H-7624 Pecs, Hungary; dora.reglodi@aok.pte.hu

**Keywords:** vascular anastomosis, vascular regeneration, microsurgery, PACAP, hemorheology, biomechanics, tensile strength

## Abstract

The proper regeneration of vessel anastomoses in microvascular surgery is crucial for surgical safety. Pituitary adenylate cyclase-activating polypeptide (PACAP) can aid healing by decreasing inflammation, apoptosis and oxidative stress. In addition to hematological and hemorheological tests, we examined the biomechanical and histological features of vascular anastomoses with or without PACAP addition and/or using a hemostatic sponge (HS). End-to-end anastomoses were established on the right femoral arteries of rats. On the 21st postoperative day, femoral arteries were surgically removed for evaluation of tensile strength and for histological and molecular biological examination. Effects of PACAP were also investigated in tissue culture in vitro to avoid the effects of PACAP degrading enzymes. Surgical trauma and PACAP absorption altered laboratory parameters; most notably, the erythrocyte deformability decreased. Arterial wall thickness showed a reduction in the presence of HS, which was compensated by PACAP in both the tunica media and adventitia in vivo. The administration of PACAP elevated these parameters in vitro. In conclusion, the application of the neuropeptide augmented elastin expression while HS reduced it, but no significant alterations were detected in collagen type I expression. Elasticity and tensile strength increased in the PACAP group, while it decreased in the HS decreased. Their combined use was beneficial for vascular regeneration.

## 1. Introduction

Microvascular anastomoses are used by several surgical specialties, most notably reconstructive and transplant surgery [1,2,3]. The development of proper vascular anastomosis is limited; therefore, there is a need to take extra care during the restoration of the blood supply of organs and tissues. Preparation of the vessels, tissue traumatization, extent of the adventitia removal, wrong geometry, bleeding, thrombosis, inflammation and ischemia-reperfusion injury can all affect the success of an anastomosis [4,5].

Tissue regeneration starts during surgery and can be divided into three main stages: inflammatory, proliferative, and remodeling. In these phases, different growth factors manage the processes [6]. Signaling between damaged and intact tissue and immune cells directly affects cell survival, tissue debris breakdown, and extracellular matrix production and maturation [6,7].

Mobilization after surgical interventions is an important part of the recovery where the increased vascular flow can play a crucial role [4]. Stretching of the vessels caused by moving can lead to damage to the anastomosis, so the tensile strength can give information about the resilience of the sutured vessels [8]. Optimal technique and the right choice of processing aids have special importance. A previous study investigated the tensile strength of end-to-end vessel anastomoses made on chicken biopreparations (~3 mm), on abdominal aortas prepared from rats post-mortem (~2 mm) and in “thread models”. They found no benefit in using simple continuous sutures instead of simple interrupted ones [9]. About the tensile strengths of the microvascular vessels and anastomoses, there are very little experimental data available, even though this is an accepted, easily reproducible, standardizable, and rapidly performed experimental method that has been repeatedly applied to the study of anastomoses of the abdominal wall or bowel [10,11,12].

Bleeding is a major complication during vascular and cardiac surgeries [13,14]. Hemostatic sponges (HS) are applied in everyday routine, and for example, Spongostan^®^ Standard (gelatine) can absorb blood up to 50 times its weight, and the whole blood clotting time is under 10 min [15,16]. The application of gelatine-type HS can modify the local signaling milieu, increase inflammation and lead to foreign body granuloma, which can all affect the tensile strength [17,18]. There have been some attempts at the impregnation of hemostatic sponges with different pharmacological agents, but none of them stimulated regeneration or decreased inflammation [19,20].

The process of microvascular anastomosis in rats and the subsequent vascular regeneration primarily involves several steps: initial surgical intervention, acute inflammation and wound healing response, re-endothelialization of the vessel wall, and maturation of the new vessel segment. These stages are orchestrated by a complex network of shearing forces, cellular processes and signaling molecules, including growth factors, cytokines, and extracellular matrix proteins [4,6,7,21]. Following anastomosis, the initial inflammatory response, driven by the recruitment of leukocytes and the release of inflammatory mediators, is crucial for clearing debris and initiating wound healing. Simultaneously, platelets play a key role in forming a thrombus that seals the wound and releases various growth factors to initiate the healing process [22]. The regeneration of the blood vessel wall, particularly neo-intima formation, is another critical aspect. Endothelial cells lining the blood vessel proliferate and migrate to cover the exposed surface, a process facilitated by vascular endothelial growth factor and other signaling molecules [23]. Over time, the new vessel segment matures, with the vascular smooth muscle cells and extracellular matrix providing structural stability to the regenerated vessel [24,25,26].

Pituitary adenylate cyclase-activating polypeptide (PACAP) is a 38-amino acid C-terminally α-amidated peptide first extracted from ovine hypothalami in 1989 [27]. PACAP is a member of the vasoactive intestinal polypeptide (VIP)-secretin-growth hormone-releasing hormone (GHRH)-glucagon superfamily, and it has three major G protein-coupled receptors, namely PAC1, VPAC1, and VPAC2. PAC1 has the highest affinity to bind PACAP, while VPAC1 and VPAC2 bind VIP and PACAP equally. Release of the neuropeptide and its receptor expression have been detected in the central nervous system (CNS) and many peripheral tissues like the respiratory tract, urinary tract, and digestive system or in bone and chondrogenic cultures [28,29,30]. Its function has been shown in vessels where PACAP induces vasorelaxation in muscular arteries such as femoral and carotid arteries [31]. Furthermore, the vasomotor effects of PACAP proved to show age dependency [32]. It can also prevent harmful cellular effects of ischemic conditions, oxidative stress, and inflammation [33,34]. PACAP has anti-apoptotic functions as well, as it has been detected in CNS, in cardiomyocytes, and in retinal ganglion cells [35,36,37,38]. The protective role and function in regeneration have been shown in bone formation, chondrogenesis, and peripheral nerve injuries [39,40,41]. In contrast, there are only sporadic data about its function in tissue regeneration after surgery, and no data can be found about its exact function in vessel healing.

Our hypothesis was that local administration of PACAP may facilitate the regeneration of end-to-end microvascular anastomoses, and the effect can be enhanced with combined PACAP and HS application. The aim of this work was to test this hypothesis in a follow-up study on the regeneration of end-to-end vascular anastomoses in rats from morphological, functional, and biomechanical points of view.

## 2. Results

### 2.1. General Observations

Respiratory parameters during anesthesia were similar in all groups; they only increased significantly (HS: *p* = 0.0216; PACAP: *p* = 0.0424) at the end of surgery (Appendix A).

Animals tolerated the topical treatment without stress and were successfully acclimatized within the first 3 days. Four animals had their arteries clotted postoperatively (two from the Control and two from the HS group). At the end of the follow-up period, we found a moderate aneurysm formation in every group at the site of the anastomosis (Control: 2; HS: 3; PACAP: 1; PACAP + HS: 2).

### 2.2. Microcirculation

Foot temperatures measured immediately before the microcirculatory measurements were almost identical in the two feet, with the highest values after the operations (Control left: 31.08 ± 2.63 °C, right: 30.47 ± 2.97 °C; HS left: 31.73 ± 0.98 °C, right: 31.74 ± 1.36 °C; PACAP left: 33.28 ± 0.96 °C, right: 33.09 ± 1.1 °C; PACAP + HS left: 32.01 ± 1.68 °C, right: 31.916 ± 1.11 °C). Except for the Control group, a decrease was detected in the other groups as the days progressed, with the lowest values measured on day 21 (Control left: 30.93 ± 0.81 °C, right: 30.95 ± 0.72 °C; HS left: 29.22 ± 1.38 °C, right: 29.84 ± 1.88 °C; PACAP left: 30.19 ± 1.27 °C, right: 30.09 ± 1.21 °C; PACAP + HS left: 29.2 ± 1.19 °C, right: 29.48 ± 0.97 °C).

The BFU values of the right and left foot were altered almost identically during the interventions. During ischemia, circulation parameters decreased in all but the operated foot of the PACAP + HS group. The postoperative value was almost identical to the values during ischemia. The lowest values for the postoperative days were as follows: Control: day 21, HS: day 14, PACAP: day 14 (left foot *p* = 0.0323), and PACAP + HS: day 21. During the postoperative days, the circulation gradually increased to maximum values by day 14 in the PACAP + HS (right foot *p* = 0.0291) and Control (left foot *p* = 0.0392; right foot *p* = 0.0004) groups. The maximum values were measured on day 7 for the HS group and day 21 for the PACAP (right foot *p* = 0.002) group (Appendix A).

### 2.3. Hematological and Hemorheological Changes

Except for the PACAP group, platelet counts were significantly increased after the surgery (Control day 14: *p* = 0.025; PACAP + HS day 7: *p* < 0.0001), and almost every group normalized on day 21. The red blood cell count decreased until day 14, after which it started to normalize. The white blood cell count increased postoperatively on day 7 in every group (Control: *p* = 0.0267; HS: *p* < 0.0326; PACAP: *p* = 0.0242; and PACAP + HS: *p* < 0.0003), most notably in the Control animals, then the values started to decrease. The PACAP-treated groups showed a slight time shift, reaching a maximum on day 14 and declining thereafter, but they were significantly different (*p* = 0.0026) from the preoperative state even on day 21 (Table 1). The hemoglobin and MCV parameters showed a continuous reduction after surgery until day 21; however, there were no significant differences between the groups. Hematocrit also decreased until day 14 and increased by day 21 (Table 1).

Red blood cell deformability showed a slight worsening on day 7 (EI_max_ in PACAP: *p* = 0.023 vs. base) and day 14 (EI_max_ in PACAP + HS: *p* = 0.0012 and SS_1/2_: *p* = 0.0039 vs. base), but on day 21, every group normalized. Red blood cell aggregation deteriorated postoperatively in every group. It was visible on the 7th postoperative day in the M 5s and M1 5s parameters, most importantly in the treated groups versus base values (M 5s in Control: *p* = 0.0278, in HS: *p* = 0.0027, in PACAP: *p* = 0.0012, in PACAP + HS: *p* < 0.0001; M1 5s in HS: *p* = 0.0008, in PACAP *p* = 0.0296), while only the PACAP group had significant (M 5s: *p* < 0.0001; M1 5s: *p* = 0.0163) differences on day 21. In the M 10s and M1 10s parameters. An elevation was observed on day 7, most notably in the HS group (M 10s in HS: *p* < 0.0001, in PACAP: *p* = 0.0078, and M1 10s in HS: *p* = 0.005 vs. base). This deterioration remained on day 21 only in the HS group (M 10s in HS: *p* = 0.0014 vs. base) (Table 2).

### 2.4. Tensile Strength

In all cases, the anastomosed femoral arteries were significantly weaker than the intact opposite arteries. Compared to the contralateral side, the smallest values were found in the HS group, while the strongest anastomoses were found in the PACAP group (Control: 0.647 ± 0.073 vs. 0.198 ± 0.089 N, HS: 0.641 ± 0.088 vs. 0.163 ± 0.059 N, PACAP 0.63 ± 0.15 vs. 0.395 ± 0.177 N, PACAP + HS: 0.694 ± 0.061 vs. 0.325 ± 0.173 N) (Figure 1A).

Ultimate tensile strength of the anastomoses increased significantly (*p* < 0.0001) decreased compared to the own intact arteries (100%): Control anastomosis was 30 ± 10.4% (*p* < 0.0001 vs. PACAP), HS had 25.9 ± 12.3% (*p* < 0.0001 vs. PACAP; *p* = 0.0456 vs. PACAP + HS), PACAP acquired 57.8 ± 8.3% (*p* = 0.0137 vs. PACAP + HS) and PACAP + HS was 39.6 ± 7.5%.

The slope of the ascending curves of tearing was reduced compared to the slope of the intact vessel. In the 33–100% section of the curves, the largest deviation, compared to intact vessels, was seen in the HS group (28.3 ± 8.9%), while the smallest deviation was in the PACAP group (64.8 ± 22.5%). The other two groups deviated almost equally (Control: 53.5 ± 13.4%, PACAP + HS 55.8 ± 16.8%) (Figure 1B).

We represented the slopes of anastomosed vessels compared to their own ipsilateral artery (100%) and all of them showed a significant difference (*p* < 0.0001): Control group was 50.3 ± 10.2% (*p* = 0.0021 vs. HS), the HS showed 24.5 ± 9%, in PACAP groups we observed 53.7 ± 13% (*p* = 0.0015 vs. HS), and in PACAP + HS, we measured 52.6 ± 10.2% (*p* = 0.0039 vs. HS).

### 2.5. Histology, Molecular Biology

#### 2.5.1. Thickness of Vessel Wall Layer

The wall thickness of the arteries showed a decrease after using HS (*p* = 0.0058). The combined PACAP + HS application was protective against the reduction. The tunica intima was not altered after wrapping with HS or PACAP in the anastomosed arteries, but a slight non-significant increase could be detected versus the Control anastomoses. The tunica media thickness showed a significant increase in PACAP + HS (*p* = 0.0121) treated anastomosed samples compared with the other anastomosed arteries. The wall thickness reduction of tunica adventitia was significant in the HS-wrapped anastomoses compared with the Control anastomoses, while PACAP-treated anastomoses also showed a slight decrease. The combined treatment reduced the decrease in the thickness of tunica adventitia in anastomosed vessels (Figure 2).

The walls of intact arteries (Control side as a benchmark) with or without PACAP treatment did not show significant alterations; furthermore, after vessel tearing, no significant changes were measured with PACAP treatments in vitro. On the contrary, PACAP induced an elevation in the wall thickness of arteries. A separate analysis of tunica intima did not show significant alterations either in intact or anastomosed vessels after PACAP treatment in vitro. A slight decrease in thickness was measured in tunica media in torn vessels, but no significant alteration was identified in tunica media in either group in vitro. Interestingly, tunica adventitia thickening was reduced by PACAP treatment in torn vessels, but a significant elevation was detected in anastomosed vessels in vitro (Figure 3).

#### 2.5.2. Elastin Expression

The elastin content of vessels was visualized and quantified with orcein staining. In the anastomosed arteries, a decreased intensity of orcein staining was detected in the HS group compared with intact and Control anastomosed vessels. On the other hand, the PACAP addition to the HS increased the orcein staining intensity in anastomosed vessels in vivo (Figure 4).

Elastin lamellas were also determined in tunica media, and some deviation was detected between the intact arteries of the different groups. In the presence of HS, a slight reduction was detected in the number of elastic rings in anastomosed tunica media. The PACAP treatment resulted in an elevation in the number of elastic lamellas in these vessels, but HS application reduced the effect of PACAP addition in anastomosed arteries. We also measured the orcein positivity of connective tissue around cannulas where PACAP treatment may diffuse. Although the elastin positivity of tissues was low around the vessels, a slight but not significant increase was determined after PACAP administration (Table 3).

The in vitro cultured vessels, without any intervention, did not show significant alteration in orcein positivity after PACAP administration, but their orcein positivity was significantly higher than in the torn vessels. In the torn vessels, PACAP administration resulted in a small but not significant reduction. On the other hand, no alteration was detected in the anastomosed vessels in the presence of PACAP. The number of elastic lamellas, similar to the in vivo experiments, was slightly increased by PACAP addition in the in vitro tissue cultures (Table 3).

Total protein expression of elastin was also detected with Western blot analysis (Appendix A). The intact and anastomosed vessels showed low expression of elastin in the Control groups. Treatments with PACAP or with HS increased the elastin expression in intact arteries and anastomosed arteries. Moreover, the PACAP addition with or without HS elevated the expression of elastin in anastomosed vessels compared with the intact arteries. Similar expression elevation was detected in PACAP-treated in vitro tissue cultures (Table 3).

Elastin immunopositivity was also detected with immunohistochemistry. In the intact vessels of the Control and HS groups, diffuse signals were detected with the appearance of some lamellated immunopositivity (Figure 5).

In the PACAP-treated groups, elastin-positive lamellas were detectable. In the PACAP-treated anastomosed vessels, diffuse and lamella-organized elastin positive signals became more prominent compared to intact vessels. The HS treatment decreased the immunopositive lamella signals of elastin in anastomosed vessels. Similar findings were detected in cultured vessels where strong diffuse elastin signals were visualized in intact in vitro cultures. The PACAP treatment increased the lamella organization compared with anastomosed in vitro artery (Appendix A).

#### 2.5.3. Collagen Type I Expression

In the tunica media of arteries, collagen type I expression influences the integrity of vessels. Therefore, the presence of collagen was investigated with picrosirius staining. In polarized light collagen, positive fibers appeared in two different colors, as thick collagen fibers show a red color and thin collagen fibers could be detected in green (Appendix A).

In intact vessels, treatment on the other side with PACAP or HS did not alter the thick collagen fibers in tunica media. The thinner collagen was lower in the intact vessels and did not alter after treatment on the anastomosed side with HS or PACAP treatment (Table 4).

In anastomosed vessels, the application of HS and PACAP significantly reduced the presence of thick collagen fibers. Furthermore, a reduction of thin fibers was also detected after PACAP and HS treatment. In contrast, the tissue’s thick and thin collagen fiber content around the cannula was elevated after PACAP administration. In the in vitro tissue cultures, we examined vessels that were torn before culturing. In this case, the PACAP treatment slightly increased the presence of thick collagen fibers without altering thin collagen fibers. In untorn benchmark arteries, similar thick collagen fibers were detected in Control groups, but a decrease was detected in PACAP-treated vessels. Moreover, the thin fibers further decreased in both Control and PACAP tissue cultures. In the anastomosed torn vessels, reduced thick fibers were detected compared to the benchmark and torn Control groups. Furthermore, a significant decrease was visible in thick and thin collagen fibers in PACAP-treated in vitro anastomosed cultures (Table 4).

Specific expression of collagen type I was further analyzed with Western blot, and similar findings were detected in vivo. The application of PACAP resulted in a reduced expression in anastomosed vessels, but no alteration was detected in the presence of HS. In the in vitro tissue cultures, the expression of collagen type I was barely detectable, but a slight elevation was detected in anastomosed cultures (Table 4).

The localization of collagen type I was also followed with immunohistochemistry (Figure 6).

The tunica media of control and treated arteries showed immunopositivity with diffuse and fiber-like structures. In the intact arteries, PACAP and HS application reduced the diffuse collagen type I signals and reduced the collagen fibers similarly to the picrosirius staining results. Moreover, the anastomosed vessels’ collagen type I immunopositivity was further reduced, and fiber signals diminished in PACAP-treated and HS groups. In the tissue cultures, diffuse and fiber signals were detected in intact vessels where the administration of PACAP further elevated the fiber content of tunica media. In cultured anastomosed vessels, collagen type I fibers were detected in tunica media. The PACAP administration elevated the diffuse collagen signals but did not augment the fiber formation (Appendix A).

#### 2.5.4. Granulomatosus Tissue around the Cannula

Pericannular granulomatous tissue was formed in all groups, but in PACAP-treated cases, it was massive. On the picrosirius-stained sections after polarization, we noticed a non-significant elevation in the PACAP-treated groups both in the red (Control and HS: 8030.26 ± 1224.92; PACAP and PACAP + HS: 17,954.93 ± 5270.92) and green (Control and HS: 3575.05 ± 2248; PACAP and PACAP + HS: 9203.76 ± 3876.22) light density. In the orcein-stained sections, there was no visible difference in the PACAP-treated groups’ (Control and HS: 1212.869 ± 183.353; PACAP and PACAP + HS: 1382.403 ± 93.97) integrated light density.

## 3. Discussion

Vascular surgeries involve various clinical factors that can lead to complications and delayed healing during anastomosis. These factors include uneven vessel diameters causing turbulent flow, longer vessel clamping leading to more cellular damage due to extended hypoxia, inadequate microcirculation and collateral blood flow. Comorbidities like hypertension, diabetes, and cancer, along with medications, might further hinder vessel healing. As a result, there is a critical need for improving the regeneration of microvascular anastomosis in clinical settings [14,42,43,44].

The effect of PACAP and/or Spongostan application on the regeneration of vascular anastomoses was investigated in this study comparatively. For the administration of PACAP, we developed a local delivery method that proved suitable for PACAP with a short half-life. Its advantages include low production cost, easy positioning and programming of the quantity and timing of dispensing, and an “inexhaustible” reservoir. Disadvantages include the environmental exposure of the dosing point, which may require animals to be kept individually, and the need for surgery in the back area if damaged. We gave 0.2 µg of PACAP to the relevant groups every second day while giving saline to the other groups. A piece of hemostatic sponge was wrapped around the anastomosis in the HS groups. The artery in the vicinity of the suture was completely encircled by the hemostatic sponge made of gelatin. This may create a mechanical barrier between the connective tissue that surrounds the anastomosis and its outermost adventitial layer. Additionally, the area may become tamponated as a result of swelling with fluid. The interaction of these may prevent various crosstalk mechanisms and fibroblast migration. In addition to aiding in suture healing, macrophages are also involved in the absorption of collagen from the HS and can cause foreign body granuloma [45,46,47].

To clarify the in vivo effects of the neuropeptide, in vitro tissue cultured vessel samples were also used to study the matrix alteration process in the presence of PACAP in a controlled manner [48]. We investigated the response mechanisms of completely intact (benchmark), freshly excised and torn intact) anastomosed vessels. Thus, the isolated vessels were not affected by crosstalk mechanisms of the surrounding tissue, inflammatory responses, or cell migration. Cultured vessels were equally exposed to PACAP, the degradation of which and its clearance by microcirculation was inhibited.

Morphological and mechanical alterations seen in the HS group may be the result of their combined involvement. In particular, PACAP can be released physiologically from nerve endings, but administration of the neuropeptide could modify these effects. After the follow-up, the HS was retained mostly in the PACAP-HS groups, where PACAP could have an impact on macrophages, influencing the healing of the vascular suture [49,50].

Regardless of the route of administration, one of the challenges of using PACAP as a therapeutic agent is its short half-life in peripheral blood due to the DPP4 enzyme. This has led researchers to investigate various methods to improve its stability and prolong its effects, including the use of PACAP analogs and delivery systems such as microspheres and nanoparticles [51] or various local administration possibilities [52]. Systemic administration of PACAP is disadvantageous due to the possible side effects and short half-life [39,53]. For animal ethical reasons and to achieve the most accurate dosing possible, repeated needle administration was not used in our study, and we developed a cannulation method similar to Larsen and Christensen [54]. As the local administration of PACAP showed protective effects in a model of dry eye disease and in neurological animal models, local use is a preferred mode of treatment [55]. For instance, local administration of PACAP-38 has neuroprotective effects in animal models of ischemic stroke, traumatic brain injury, and Parkinson’s disease [56,57].

As the result of precisely positioned neuropeptide addition, the laboratory parameters did not show specific alterations regarding the systemic effect of PACAP. However, in the case of white blood cell count, the anti-inflammatory effect of PACAP was observed in the first week, with an increase in the later days, which may be related to the absorption disturbances following the formation of the tissue channel. Red blood cell aggregation showed a deterioration in the postoperative weeks compared to the Control group. The PACAP group showed the greatest deterioration on day 21. It is known that inflammatory processes, acute phase reactions, free radical reactions and metabolic changes may alter micro-rheological parameters, such as red blood cell deformability and aggregation [58,59,60].

The tensile strength of anastomoses was expressed in newtons in pair with the contralateral intact arteries according to the different groups. An intact artery has a tensile strength of 0.652 ± 0.094 N, whereas the single strand of suture material we used measured 0.522 ± 0.047 N. Since the tensile strength of the eight knots (16 threads of suture material) used simultaneously was greater than the tensile strength of the intact artery, we did not examine the tearing or unbinding of the suture material. In most cases, the material will rip at its weakest point, which is the damaged, anastomosed and partially regenerated area in our case. Our observations, in which the sutures failed in every instance close to the anastomoses, are consistent with this. Therefore, we can compare the data obtained that are only affected by the actual regeneration of the anastomoses [61,62].

Tensile strength values of the anastomosed arteries were significantly lower on day 21 compared to intact arteries. PACAP group had the strongest anastomoses, while the HS and Control anastomoses seemed the weakest compared to the anastomosed arteries. The slope of the curves can refer to the elasticity of the arteries. More rigid tissues have a higher slope, so they can reach the maximum point faster than the elastic tissues with smaller slopes [63]. The results show that the vessel rupture pattern is similar to that of elastomers [64,65]. In the first third of the curve, the slope is typically lower, followed by a gradually parabolically rising curve. Because of the irregularity of the first third of the slopes, we only analyzed the slope of the curve between 33–100% of the whole curve. It was lower in every anastomosis compared to intact arteries. The discrepancy with the intact arteries in the tensile strength and slopes reflects partial regeneration processes on the 21st day, which seemed the worst in the HS group, probably caused by the inappropriate maturation [66,67,68,69,70].

We compared the general morphological differences and measured the thickness of the tunica intima, media, and adventitia in H&E-stained vessels. In histological sections of freshly excised and torn vessels, the thickness of the layers showed a relatively big standard deviation. In the tensile strength test, the tissue layers were likely to have been disrupted and stretched, which could have reacted differently to formalin. Tunica intima of arteries did not alter in the groups. Tunica media also showed some similarities, but it significantly increased in the PACAP + HS group. In Control anastomoses, tunica adventitia thickened, while in the HS group, it significantly decreased. In the PACAP group, there was a non-significant decrease, while the combined PACAP + HS treatment led to an antagonistic effect. These alterations in vessel structure can be the result of multiple signaling activation by PACAP, as it has been published that PACAP can inhibit angiogenesis and fibroblast accumulation [71]. It is also able to decrease matrix-degrading enzyme functions such as matrix metalloproteinase-2 and -9 [41,71], which can prevent degradation of the tunica media and adventitia. Subsequently, the continuous presence of PACAP with HS resulted in a better and slower signaling activation, keeping the vessel sublayer reconstruction in balance during regeneration. A single application of HS can keep the sutures and the vessels more intact [17]. However, it was not able to activate protecting signaling pathways, which can induce vessel regeneration or extracellular matrix expression, as PACAP did in callus formation [40] or in chondrogenesis [33].

In tissue-cultured arteries, the cells were able to survive the mechanical trauma after tensile strength measurement, and there was time for the restoration of the wall structure, so there were less apparent variations in the results. In the anastomosed and intact arteries, the torn trauma was prevented by PACAP treatment, which was also visible in histological images; in addition, an increase in the thickness of the layers in the anastomosed arteries was identified. The effect of PACAP treatment on the extracellular matrix was also seen in the un-ruptured benchmark arteries, but these differences were not significant. PACAP administration in vitro elevated the thickness of the layer in the non-damaged benchmark and, for the second time, damaged anastomosed arteries. This provides further evidence that PACAP can induce the expression of extracellular matrix components such as collagen type I in cultured conditions, as was shown in osteoblasts [30,72]. For the intact and freshly damaged arteries, PACAP lowered the layer thickness, which also suggests that PACAP can prevent the harmful effect of strong mechanical forces until a certain threshold and in the lack of a complete system, it is not able to reverse the degrading processes [29,34]. Although this study proved the unquestionable effects of PACAP on vessel regeneration, the controversial results between the in vivo and in vitro experiments may be due to the presence and absence of PACAP degrading enzymes in vivo and in vitro, respectively, leading to different local effects of the neuropeptide.

The arrangement of elastic fibers was examined by orcein and immunohistochemical stainings. The control group had the strongest staining, and the HS group had the least staining. In the PACAP + HS group, we also could see some antagonistic effects. The internal and external elastic laminas were well observed on the slides. In the tunica media, there were more membrane-organized elastic lamellas observed in the PACAP group, however, PACAP treatment could not protect against the HS-lowered number membranes. PACAP also increased the elastin content of the pericannular connective tissue in vivo. With immunohistochemistry, increased immunopositivity of elastic fibers was visualized in the tunica media after PACAP treatment. A stronger diffuse positivity was also visible due to newly formed elastic membranes. Spongostan gelatin-based biomaterials have specific elasticity, maintaining the normal function of sutured vessels and triggering better wound healing [17] without affecting extracellular matrix production and organization. While PACAP can induce matrix formation until a well-balanced level, as it was shown in collagen type IV expression in the kidney [73], we do not have data on its direct function on elastin formation. This is the first study to show the role of neuropeptides in elastin expression and organization in vessel regeneration. Furthermore, in vitro PACAP could increase elastin expression in benchmark and anastomosed arteries while the number of elastic membranes slightly increased in every group.

Western blot analysis showed increased expression of the elastic fibers in the PACAP group compared to the Control anastomoses and intact arteries. These results further support the direct effects of PACAP on elastin expression without systemic activation, but elastin lamellar organization can be partly regulated by mechanical activation, such as vasorelaxation or vasodilation. Therefore, HS application may reduce the mechanical movement of vessels, which can be partially the reason for decreased elastin expression and lamella formation.

With picrosirius red staining, we saw non-specific collagen labeling with red coloring. By the rotation of polarized light with λ/4, we can evaluate the orientation of collagen type I fibers and their thickness. The green light intensity belongs to the thinner fibers, while the red color intensity belongs to the thicker fibers. There were no significant differences between the anastomoses, but we could see a decrease in the thick fibers and an increase in the thin fibers in the HS group. PACAP treatment resulted in fiber thickness similar to that of the Control arteries. In bone formation and regeneration, PACAP was demonstrated to increase collagen type I production to a certain threshold [30,40], and it has also been published that the presence of the neuropeptide can inhibit the activation of matrix-degrading enzymes such as hyaluronidases and matrix metalloproteinases [41]. Therefore, the protective effect of PACAP in vessel regeneration seems to be rather via preventing collagen degradation than collagen production. On the other hand, collagen type I immunohistochemistry yielded signals in the tunica media, suggesting that the collagen production of the cells here is modified by the neuropeptide. Therefore, it is likely that PACAP increased collagen production in the tunica media in an area-specific manner but did not result in fibrotic accumulation in the outermost layer due to its antifibrotic function [74,75]. On the contrary, in our experiment, the microsurgical preparation of the PACAP-treated anastomoses was more difficult on the 21st postoperative day. We found massive pericannular tissue in which the content of thick and thin collagen fibers was elevated. This finding points out that the application of PACAP with foreign bodies, like cannulas and/or implants, may alter the physiological process of PACAP-activated signaling pathways. Similar results were shown in some pathologically altered tissue formations, such as tumors where PACAP and its receptors were overexpressed and triggered tumor progression [76].

In summary, PACAP treatments could facilitate the depletion and maturation of the extracellular matrix. In the tensile strength test, the slope of the force-time curves decreased, expressing better elasticity. The peak of the curves (tensile strength) was the highest in PACAP-treated groups and the lowest in the HS group, supposedly due to the barrier effect of the topical sponge. The presence of the HS could decrease cell migration and compress the vasa vasorum. Moreover, PACAP primarily affected the media and resulted in elastin expression and organization without modifying collagen expression.

Limitations of the study include the relatively low case number, the possible inter-species differences, and the supposed crosstalk between the externally administered PACAP and the endogenous mechanisms. We have chosen gelatin HS, but it is also supposed that other topical agents might have different effects. The PACAP dosage and administration timing should also be investigated further.

## 4. Materials and Methods

### 4.1. Experimental Animals and Drugs Used

The experiment was registered and approved by the University of Debrecen Committee of Animal Welfare and by the National Food Chain Safety Office (registration Nr. 25/2016/UDCAW) following the national (Act XXVIII of 1998 on the Protection and Sparing of Animals) and EU (Directive 2010/63/EU) regulations.

Thirty-four Wistar male rats (321.23 ± 37.1 g, Toxi-Coop Zrt., Budapest, Hungary) had been subjected to this study. Animals arrived at the Department’s conventional animal house at the age of six weeks, where they were acclimatized for one and a half weeks. During the preparation, the animals were weighed and, depending on their body weight, were given general anesthesia with a specific combination of Ketamine-Xylazine-Atropin (100 mg/kg; 10 mg/kg; 0.05 mg/kg) [77]. As prophylaxis for thrombosis, heparin was administered via the lateral tail vein (80 IU/kg). For postoperative analgesia, conventional NSAIDs were excluded to evaluate the healing process, and therefore, Tramadol (15 mg/bwkg/day) was administered intraperitoneally [78] at the end of surgery and on the first 3 p.o. days.

### 4.2. Operative Techniques, Experimental Groups and Sampling Protocol

A blunt-tipped 22G 1× (Neoject, Dispomed GmbH&Co. KG, Gelnhausen, Germany) needle was inserted into a 20-cm long polyethylene (Polyethylene Tubing, Clay Adams, 427411, BD Intramedic™, Sollentuna, Sweden) tube (0.965 mm outer diameters, approximately 0.05 mL volume). This was cut to the size of each animal during surgery, considering that animals often rest curled up. To ensure fixation, we drilled through the stiffening wings on both sides of the needle with a 2 mm drill bit. After preparation, we disinfected the inside and outside with betadine and washed them with physiological saline before insertion.

The surgical sites (between the two scapulae and on the right thigh) were shaved and disinfected with betadine, and an incision of about 4 cm was made in the right posterior inner thigh, below the level of the inguinal ligament and parallel to it. The underlying fatty tissue was cut parallel to the abdominal wall with scissors along the length of the incision and then pulled distally using a small hemostatic clamp. The fatty connective tissue around the artery was prepared, and the artery was isolated from the level of the inguinal ligament to the origin of the saphenous and popliteal arteries, separated from the femoral vein and nerves. A safety clip was placed on the artery under the inguinal ligament, followed by one jaw of the approximator, pulled to the lateral position, and the other jaw placed in front of the distal division of the femoral artery. Halfway between the two, the vessel was transected in a standardized manner. Washing with heparin (~0.4 mL, 2500 IU/mL) followed, and to inhibit absorption from the surgical site, we used gauze to absorb the excess and then washed the site of the invasion with physiological saline. The minimal amount of adventitia required for stitches was removed. Sutures were prepared using polyamide-6 monofilament (Daclon, SMI, Vith, Belgium) suture material containing a 10/0 serosa needle, and eight single-knot sutures were prepared in each case (Figure 7A). The patency of the anastomosis was checked in all cases with a “milking test” [79].

From the inguinal region, a channel was prepared with curved Kocher subcutaneously through the back of the animal to the shaved area between the shoulder blades, where an incision was done and the cannula was inserted. Above and below the needle, we closed the wound with Donati stitches (4/0 polyglycolic acid, Surgicryl Rapid, SMI, Vith, Belgium) and secured the needle with two single swing knots (3/0 Silk, braided, SMI, Vith, Belgium). The other end of the cannula was passed over the vessel (~1 cm) and secured at three points in the surrounding tissue (8/0 Polyamide-6, monofilament, Vitrex, Herlev, Denmark) (Figure 7B). Finally, a hole (1.05 ± 0.034 mm) was cut in the cannula just above the anastomosis for local treatment. Then, the wound edges were wiped with Betadine gauze, and a horizontal mattress suture (4/0 polyglycolic acid, Surgicryl Rapid, SMI, Vith, Belgium) was used to close the surgical site.

The animals were randomly subjected to Control (n = 8), HS (n = 8), PACAP (n = 8) and PACAP + HS (n = 8) groups, according to the treatments.

The PACAP groups received 0.2 µg PACAP 1-38 (100 μg/vial, provided by Prof. Gabor Toth from the University of Szeged) every two days (first treatment immediately after surgery) dissolved in 0.2 mL physiological saline, to which was added a further 0.2 mL physiological saline. The other groups also received 0.4 mL of physiological saline every two days via the dosing point. During the formation of the HS groups, a 3 × 4 mm piece of Spongostan Standard™ (Raritan, Franklin Township, NJ, USA) hemostatic sponge was wrapped around the anastomosis (Figure 7C). For the PACAP + HS group, the first dose of PACAP was injected directly into the sponge at the end of the surgery.

Pre-operatively, and on days 7, 14, and 21 postoperatively, a lateral tail vein was cannulated (26 G) through which 0.6 mL of blood was drawn (K3-EDTA, Vacutainer^®^, Becton Dickinson GmbH, Franklin Lakes, NJ, USA), followed by 1 mL of physiological saline, and before removing the cannula at the end of the surgery, an additional 1 mL was given as a fluid replacement.

During the follow-up period, regular wound care was taken. At the end of the 3-week follow-up period, we took the specific samples, and the animals were exterminated. The anastomosed and intact contralateral vessels were dissected from the internal iliac branch to the popliteal branching. The vessels were clipped as proximal as possible immediately before measurement. The blood vessels were tested as fresh as possible, so the animals were sacrificed only after the blood vessels had been sampled. After both blood vessels were removed, the animal was given an intravenous Ketamine-Xylazine solution (50 mg/bwkg; 5 mg/bwkg) for the termination. We gave it only at the very end to prevent the highly concentrated solutions from affecting the vessels.

The vessel samples were prepared for the histological, cell culture and tensile-strength measurements and analyses, the protocol of which is summarized in Figure 8.

### 4.3. Laboratory Tests

The hematological measurements were performed by a Sysmex K-4500 automated hematology analyzer (TOA Medicor Electronics Co., Ltd., Tokyo, Minato City, Japan). In this study, red blood cell count (RBC [10^12^/L]), white blood cell count (WBC [10^9^/L]), hematocrit (Hct [%]), hemoglobin concentration (Hgb [g/L]), mean corpuscular volume (MCV [fL]), and the platelet count (Plt [10^9^/L]) were analyzed.

A LoRRca MaxSis Osmoscan ektacytometer (RR Mechatronics International BV, Zwaag, The Netherlands) was used to assess erythrocyte deformability [80,81]. Polyvinyl-pyrrolidone (PVP) and the phosphate-buffered saline (PBS) solution (viscosity: 28 mPas, osmolarity: 294 mOsm/kg, pH: 7.3) were diluted with 10 µL of blood. The red blood cell deformability was expressed by the elongation index (EI) in the function of shear stress (SS [Pa]). The Lineweaver–Burk equation was used to calculate the maximal elongation index (EI_max_) and the shear stress at half EI_max_ (SS_1/2_ [Pa]) from the individual EI-SS curves [82].

Red blood cell aggregation was tested using the method based on light transmission (Myrenne MA-1 aggregometer, Myrenne GmbH, Roetgen, Germany). Four aggregation parameters were determined using 20 µL blood/measurement: aggregation M index values under stasis (M 5s, M 10s) and M1 values at 3 s^−1^ shear rate (M1 5s, M1 10s) [80,81].

### 4.4. Microcirculatory Measurements

Microcirculatory measurements were performed before surgery, 3 min after arterial occlusion, at the end of the surgery, on days 7, 14, and 21 on the lower base of the distal footpad on the lateral side of the feet of the rats. We used an LD-01 laser Doppler flowmeter (Experimetria Co., Budapest, Hungary) with a standard pencil probe (MNP100XP, Oxford Optronix Ltd., Adderbury, UK). This machine provides a relative parameter (blood flux unit, BFU [au]) about the microcirculatory of the investigated tissue area (cca. 1 mm^3^), which is integral over the velocity and number of moving red blood cells [83].

A 20-second BFU recording part was averaged in all the measurements. We also checked the skin temperature (ri-thermo^®^ N professional Clinical Thermometer, Riester, Phoenix, AZ, USA) at the microcirculatory measurement sites.

### 4.5. Tensile Strength Measurement of the Anastomoses

To measure the tensile strength, we used an upgraded version of a custom-made device in cooperation with the Department of Information Technology, Faculty of Informatics and the Department of Operative Techniques and Surgical Research, Faculty of Medicine, University of Debrecen, Hungary [9,84]. This developed version contains a new aluminum frame, an Arduino Mega 2560 Rev3 microcontroller board, and a 72 W power supply module (RS-75-12, MEAN WELL Enterprises Co., Ltd., New Taipei City, Taiwan).

A DRV882 integrated motor driver guides the NEMA17 stepper motor (42BYGHW811M, Kuongshun Electronic Ltd., Shenzhen, China) attached to an HPV7 C-Beam Linear Actuator (Guangzhou Hanpose 3D Technology Co., Ltd., Guangzhou, China). One of the holding clamps (HJJ-001, Baoshishan, Liaoning Sheng, China) is attached to the Z axle sliding table, while the other clamp is fixed to a weight sensor (TAL220, max. 5 kg, Deecom Technology, Kuala Lumpur, Malaysia). The analog signal from the load cell is digitalized with a weighing sensor 24-bit A/D conversion adapter (HX711, Avia Semiconductor (HX711, Avia Semiconductor Xiamen Ltd., Xiamen, China). The machine is connected to a computer via a USB 2.0 port and communicates with a universal serial monitor (QtSerialMonitor 1.5 by Michal W., open-source software on GitHub Inc., San Francisco, CA, USA).

The removed arteries were always fixed between the same place in the clamping jaws, 1 cm apart, and the anastomosis point was centered. The pulling force made by the motor (48.66 steps/s; 1.95 mm/s) was registered in grams, and data was exported into a CSV file. For the data management, we also made the gram/newton conversion (9.81 m/s^2^), and the maximum (rupture point) and slope of the force-time curves were analyzed (Figure 9).

The analysis was performed on the range of curves above 0.0098 N, which, according to the accuracy of the instrument, no longer contained any irregular parts. Since the curves had a similar rupture pattern to elastomers, the initial more irregular one-third was ignored, and only the slope of the ascending section (33–100% of the whole tensile strength curve) was analyzed (tgα), which provides information on the integrity and elasticity of the collagen that primarily provides mechanical stability to the blood vessels [64,65].

### 4.6. Tissue Cultures

Based on the culture procedure finalized during the pre-experiment in the tissue culture room, a portion of the severed blood vessels was cultured in HG-DMEM medium (Dulbecco’s modified eagle medium (Sigma-Aldrich, St. Louis, MO, USA) and 10% FCS (fetal calf serum; Gibco, Gaithersburg, MD, USA) supplemented with antimicrobial agents (penicillin, streptomycin, ampicillin, fungizone, + ascorbic acid) are placed in Petri dishes (Eppendorf, Hamburg, Germany), incubated at 37 °C in a 5% CO_2_ and 95% humidity and cultured for 14 days. During culturing, some samples were treated with PACAP (10 μL PACAP, 10 nmol/L concentration) every 48 h according to the time of medium exchange according to the groups. These arteries removed from animals were originally from the control or treated groups in vivo.

Besides the intact and anastomosed arteries, we also tested PACAP on a group (intact as benchmark arteries) of intact, freshly prepared arteries without any mechanical trauma. From the two remaining rats, we prepared four femoral arteries and then cut them in half with a microsurgical scissor. These artery pieces were similar in size to the arteries torn into two pieces (approximately 1 cm). For the experiments, equally proximal and distal parts of vessels were used. During the in vitro experiments, we continued the same treatments on certain vessels as in vivo, except for the benchmark group, which was not torn and treated in vivo.

### 4.7. Light Microscopical Morphology

Vessels were washed in PBS three times and fixed in a 4:1 mixture of absolute ethanol and 40% formaldehyde, then embedded in paraffin. Serial sections were made, and haematoxylin-eosin staining (H&E, Sigma-Aldrich, St. Louis, MO, USA) for morphological analysis and orcein staining (Sigma-Aldrich, St. Louis, MO, USA) for elastin visualization were performed. Staining protocols were carried out according to the manufacturer’s instructions. Photomicrographs were taken using a DP74 camera (Olympus Corporation, Tokyo, Japan) on an Olympus Bx53 microscope (Olympus Corporation, Tokyo, Japan). For the examination of the orientation of collagen fibers in vessels Picrosirius red (Sigma-Aldrich, St. Louis, MO, USA) staining was used. In polarized light, turning the light plane with λ/4 samples was investigated in an Olympus Bx53 polarization microscope (Olympus Corporation, Tokyo, Japan).

For the measurement of tunica intima, media, and adventitia thickness, 20× magnification photomicrographs of H&E staining were investigated with ImageJ 1.40 g freeware. We drew a perpendicular line from the internal elastic membrane to the tunica adventitia, and the pixel numbers were determined. Results are shown as a percentage of control pixel numbers. Twenty independent measurements were made in one slide, and four independent arteries were used per experimental group. In orcein staining, the normal color is inverted to green and black to increase the contrast of the pixels. The green pixel number was determined by ImageJ software. An equal area of the vessel wall of 20× objective magnification was used to determine orcein (green) positivity and was shown in graphs.

### 4.8. Immunohistochemistry

In vessels, localization of collagen type I and elastin were visualized. Vessels were fixed in a 4:1 mixture of absolute ethanol and 40% formaldehyde and washed in 70% ethanol. After embedding serial sections were made, deparaffination was then followed by rinsing in PBS (pH 7.4). Non-specific binding sites were blocked with PBS supplemented with 1% bovine serum albumin (Amresco LLC, Solon, OH, USA), then samples were incubated with monoclonal collagen type I (Sigma-Aldrich, St. Louis, MO, USA) at a dilution of 1:500 and polyclonal elastin (Sigma-Aldrich, St. Louis, MO, USA) at a dilution of 1:1500 at 4 °C overnight. For visualization of the primary antibodies, the anti-rabbit Alexa Fluor 555 secondary antibody (Life Technologies Corporation, Carlsbad, CA, USA) was used at a dilution of 1:1000.

Samples were mounted in Vectashield mounting medium (Vector Laboratories, Peterborough, UK) containing DAPI for nuclear DNA staining. For negative controls, anti-rabbit Alexa Fluor 555 was used without the primary antibodies. Photomicrographs were taken using a DP74 camera (Olympus Corporation, Tokyo, Japan) on an Olympus Bx53 microscope (Olympus Corporation, Tokyo, Japan). Images were acquired using Cell Sense Entry 1.5 software (Olympus, Shinjuku, Tokyo, Japan) with constant camera settings to allow the comparison of fluorescent signal intensities. Images of Alexa555 and DAPI were overlaid using Adobe Photoshop version 10.0 software. The contrast of images was equally increased without changing constant settings.

### 4.9. Western Blot Analysis

Vessel samples were washed in physiological saline and stored at −70 °C. Samples were mechanically disintegrated with a tissue grinder in liquid nitrogen. Then they were collected in 100 μL of homogenization RIPA (Radio Immuno Precipitation Assay)-buffer (150 mM sodium chloride; 1.0% NP40, 0.5% sodium deoxycholate; 50 mM Tris, pH 8.0) containing protease inhibitors (Aprotinin (10 μg/mL), 5 mM Benzamidine, Leupeptin (10 μg/mL), Trypsin inhibitor (10 μg/mL), 1 mM PMSF, 5 mM EDTA, 1 mM EGTA, 8 mM Na-Fluoride, and 1 mM Na-orthovanadate). The suspensions were sonicated by pulsing burst for 30 s at 40 A (Cole-Parmer, Vernon Hills, IL, USA). Total cell lysates for Western blot analyses were prepared. An amount of 20 μg protein was separated in 7.5% SDS–polyacrylamide gels for the detection of collagen type I, elastin, and actin. Proteins were transferred electrophoretically to nitrocellulose membranes and exposed to the primary antibodies overnight at 4 °C in the dilution given above. After washing for 30 minutes with PBST, membranes were incubated with the peroxidase-conjugated secondary antibody anti-rabbit IgG in a 1:1500 (Bio-Rad Laboratories, Hercules, CA, USA) or anti-mouse IgG in 1:1500 (Bio-Rad Laboratories, Hercules, CA, USA) dilution. Signals were detected with enhanced chemiluminescence (Advansta Inc., Menlo Park, CA, USA) according to the instructions of the manufacturer. We used actin as a housekeeping protein (internal control), and the results of elastin and collagen type I expression were normalized on the expression of actin. Signals were developed with a gel documentary system (Fluorchem E, ProteinSimple, San Jose, CA, USA). Optical densities of signals were measured by using ImageJ 1.40 g freeware. Statistical analysis between the experimental groups was determined based on the result of the data normality test.

### 4.10. Statistical Analysis

Mead’s resource equation method was used to estimate the necessary sample size (animals per group) [85]. GraphPad Prism 9.1.2 (226) software was used for statistical analysis. Normality was checked for all data distributions, and accordingly, Student *t*-test or non-parametric tests (Wilcoxon or Mann-Whitney rank sum tests) and two-way/repeated measures ANOVA tests were used. The significance level was set at *p* < 0.05.

## 5. Conclusions

Both the administration of PACAP and the application of HS had a beneficial effect on the vascular regeneration process. The in vitro and in vivo patterns were not always consistent for vessels of different conditions (intact or anastomosed), which suggests the importance of different local factors, hormones, mechanical activation, and prior interventions like the extent of tissue traumatization. HS primarily impaired the mechanical properties of the vessels, while administration of PACAP seems to be a promising agent in vascular regeneration that requires further investigation. Hematological and hemorheological alterations were non-specific for groups; however, an uncertain anti-inflammatory effect was seen, suggesting a systemic influence of the PACAP.

## Figures and Tables

**Figure 1 ijms-24-16695-f001:**
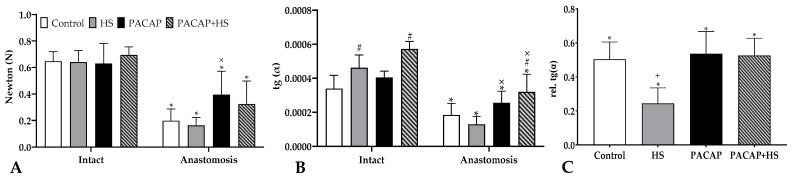
The maximal tensile strength values needed for the tearing of the vessels in newton (**A**), the slope in the 33–100% range of the full slopes of the tensile strength graphs (**B**), and the relative changes in slope values compared to base (**C**). Mean ± S.D. * *p* < 0.05 vs. Intact artery (control side), # *p* < 0.05 vs. Control group, × *p* < 0.05 vs. HS group, + *p* < 0.05 vs. all other groups.

**Figure 2 ijms-24-16695-f002:**
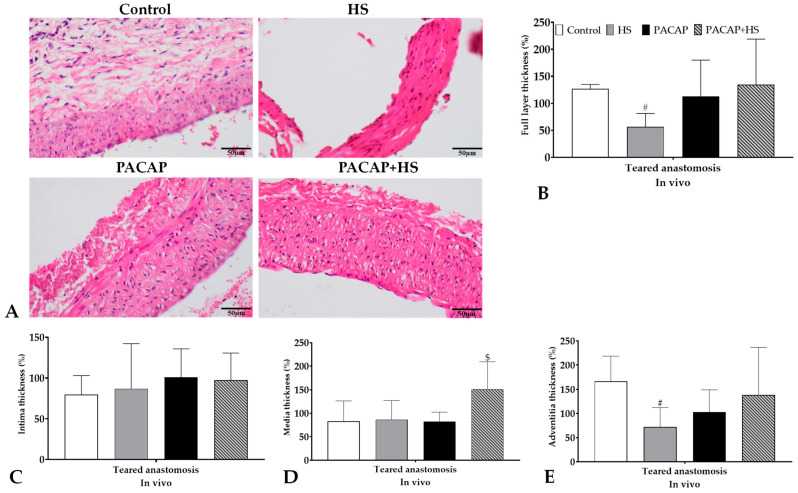
Representative histological photos (original magnification: 20×) and vessel wall layer thickness data of the excised and torn (by tensile strength test) arteries. (**A**) Representative H&E-stained slides; (**B**) full layer thickness; (**C**) tunica intima thickness; (**D**) tunica media thickness; and (**E**) tunica adventitia thickness. Means ± S.D., # *p* < 0.05 vs. Control, $ *p* < 0.05 vs. all other groups.

**Figure 3 ijms-24-16695-f003:**
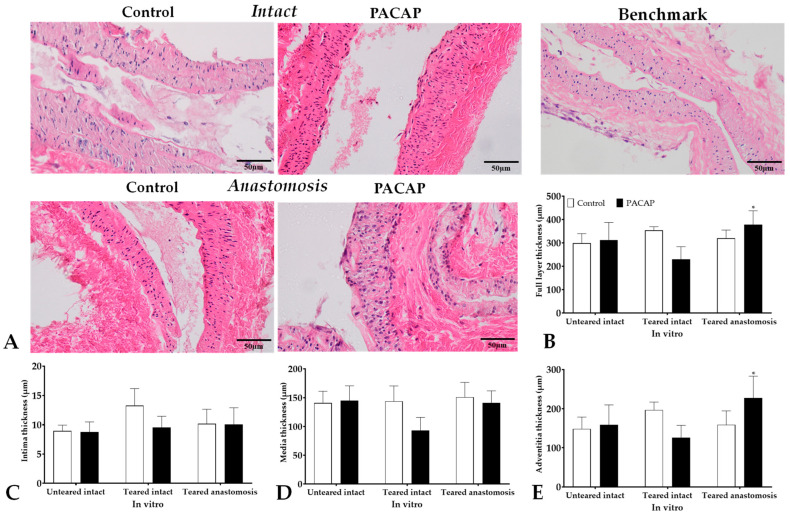
Representative histological photos (original magnification: 20×) and vessel wall layer thickness data of the in vitro cultured (14 days after tearing: intact or anastomosis; 14 days after harvesting from a rat: Control benchmark) arteries. (**A**) Representative H&E-stained slides; (**B**) full layer thickness; (**C**) tunica intima thickness; (**D**) tunica media thickness; and (**E**) tunica adventitia thickness. Means ± S.D., * *p* < 0.05 vs. Control.

**Figure 4 ijms-24-16695-f004:**
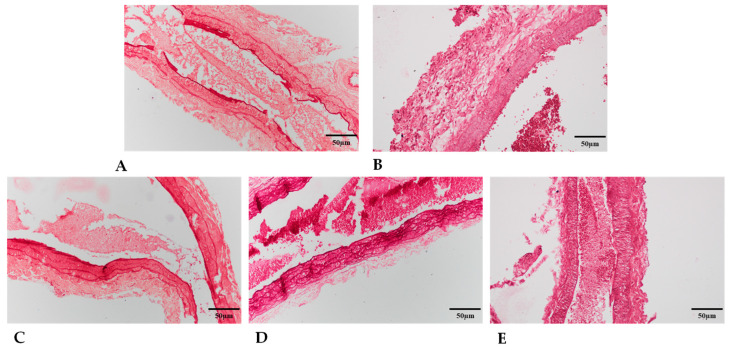
Representative orcein stained histological photos (original magnification: 20×) of the intact Control artery (**A**); anastomosed arteries in Control (**B**); HS (**C**); PACAP (**D**); and PACAP + HS (**E**) groups.

**Figure 5 ijms-24-16695-f005:**
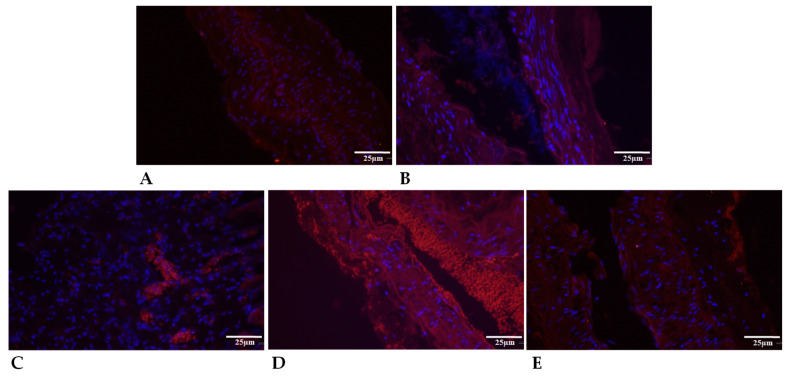
Representative pictures for elastin immunohistochemistry (original magnification: 40×) of intact Control artery (**A**); anastomosed arteries in Control (**B**); HS (**C**); PACAP (**D**); and PACAP + HS (**E**) groups. Blue: DAPI; Red: elastin antibody-positivity.

**Figure 6 ijms-24-16695-f006:**
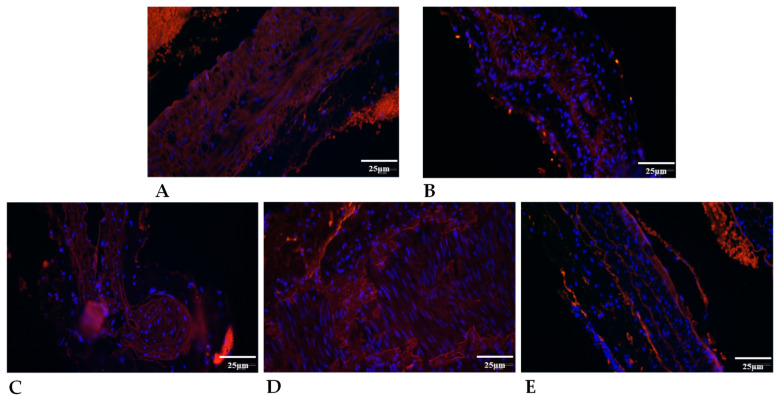
Representative pictures for collagen type I immunohistochemistry (original magnification 40×) of intact control artery (**A**); anastomosed arteries in Control (**B**); HS (**C**); PACAP (**D**); and PACAP + HS (**E**) groups. Blue: DAPI; Red: collagen type I antibody-positivity.

**Figure 7 ijms-24-16695-f007:**
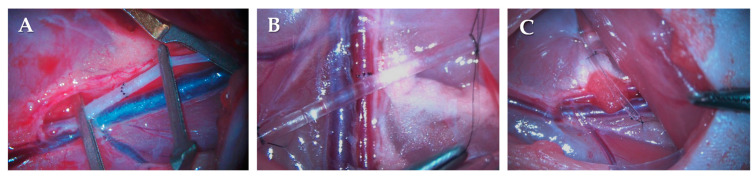
Intraoperative photos of the finished end-to-end anastomosis on the femoral artery (**A**); positioning the cannula for administration of drugs (**B**); and the applied Spongostan HS piece wrapping the anastomosis line with the cannula placed next to it (**C**).

**Figure 8 ijms-24-16695-f008:**
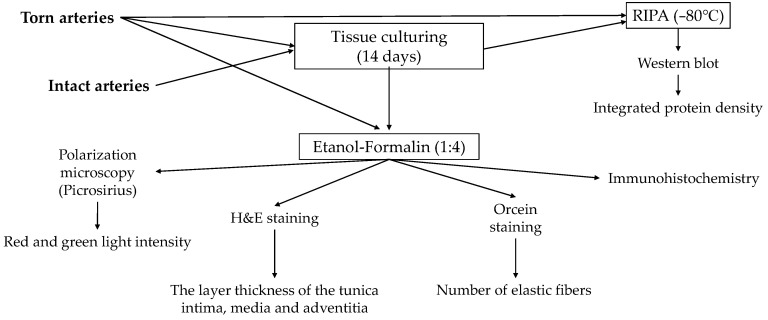
Schematics of the protocol for tissue sample preparation and procedures.

**Figure 9 ijms-24-16695-f009:**
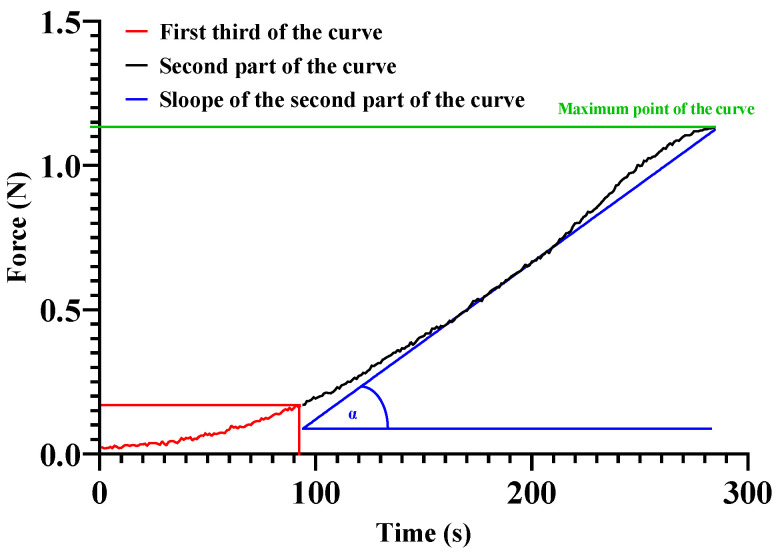
Representative tensile strength measurements force-time curve and the analyzed parameters. The first one-third of the curve (red) was not included in the slope calculation due to its irregularity.

**Table 1 ijms-24-16695-t001:** Alterations of the selected hematological parameters (white blood cell count—WBC; red blood cell count—RBC; hematocrit—Hct; hemoglobin concentration—Hgb; mean corpuscular volume—MCV; platelet count—Plt) in the Control, HS, PACAP and PACAP + HS groups during the follow-up period.

Variable	Group	Base	7th p.o. Day	14th p.o. Day	21st P.O. Day
WBC [10^9^/L]	Control	11.95 ± 3.63	18.92 ± 7.94 *	18.89 ± 3.18 *	13.82 ± 4.84
	HS	10.88 ± 40	17.47 ± 6.37 *	17.79 ± 3.15 *	15.88 ± 5.07
	PACAP	11.11 ± 4.65	16.31 ± 2.43 *	21.95 ± 6.78 *	18.81 ± 5.67 *
	PACAP + HS	10.04 ± 3.58	15.29 ± 4.04 *	16.13 ± 4.12 *&	13.38 ± 2.72 *&
RBC [10^12^/L]	Control	7.59 ± 0.48	6.84 ± 0.81	7.02 ± 0.34 *	7.44 ± 0.37
	HS	7.57 ± 0.44	7.51 ± 0.34 #	7.35 ± 0.29 #	7.22 ± 0.61 *
	PACAP	7.81 ± 0.34	7.63 ± 0.33 #	7.29 ± 0.49 *	7.63 ± 0.21
	PACAP + HS	8.01 ± 0.34 +	7.33 ± 0.55 *	7.40 ± 0.23 *#	7.50 ± 0.46 *
Hct [%]	Control	44.66 ± 2.79	39.23 ± 3.05 *	39.75 ± 1.92 *	41.76 ± 2.26 *
	HS	44.38 ± 1.8	42.34 ± 2.40 *#	41.06 ± 2.35 *	40.06 ± 3.50 *
	PACAP	45.60 ± 1.85	43.16 ± 2.52 *#	40.54 ± 2.26 *	42.19 ± 2.28 *
	PACAP + HS	46.15 ± 1.87	40.75 ± 2.96 *	41.04 ± 1.58 *	41.54 ± 2.20 *
Hgb [g/L]	Control	14.9 ± 1.01	13.35 ± 1.43 *	13.14 ± 0.75 *	13.61 ± 0.81 *
	HS	15.09 ± 0.69	14.34 ± 0.88 *	13.51 ± 0.76 *	12.96 ± 1.07 *
	PACAP	15.40 ± 0.61	14.56 ± 0.92 *	13.38 ± 0.84 *	13.70 ± 0.87 *
	PACAP + HS	15.53 ± 0.45	13.71 ± 0.98 *	13.43 ± 0.41 *	13.36 ± 0.81 *
MCV [fL]	Control	58.90 ± 2.13	57.76 ± 4.59	56.69 ± 2.77 *	56.19 ± 1.78 *
	HS	58.73 ± 1.69	56.33 ± 2.04 *	55.82 ± 2.09 *	55.56 ± 1.89 *
	PACAP	58.41 ± 1.44	56.56 ± 1.73 *	55.70 ± 1.76 *	55.24 ± 2.07 *
	PACAP + HS	57.51 ± 10	55.59 ± 1.08 *	55.48 ± 0.98 *	55.43 ± 1.11 *
Plt [10^9^/L]	Control	718.43 ± 236.20	900 ± 140.97	990.93 ± 181.6 *	835.86 ± 129.69
	HS	647.14 ± 209.38	877.93 ± 232.37	719.79 ± 375.57	598.93 ± 216.68 #
	PACAP	736.50 ± 115.93	899.63 ± 254.41	819.81 ± 266.16	751.06 ± 196.66
	PACAP + HS	682.19 ± 220.62	1025.63 ± 173.21 *	1031.31 ± 236.01 *	917.31 ± 135.92 *+&

Means ± S.D., * *p* < 0.05 vs. base, # *p* < 0.05 vs. Control, + *p* < 0.05 vs. HS, & *p* < 0.05 vs. PACAP.

**Table 2 ijms-24-16695-t002:** Changes in red blood cell deformability (EI_max_ and SS_1/2_) and red blood cell aggregation parameters (aggregation indices M 5s, M 10s, M1 5s, and M1 10s) in the Control, HS, PACAP, and PACVAP + HS groups during the follow-up period.

Variable	Group	Base	7th p.o. Day	14th p.o. Day	21st p.o. Day
EI_max_	Control	0.57 ± 0.01	0.55 ± 0.03	0.57 ± 0.01	0.58 ± 0.02
	HS	0.58 ± 0.02	0.57 ± 0.01	0.58 ± 0.01	0.58 ± 0.02
	PACAP	0.58 ± 0.02	0.57 ± 0.02 *	0.57 ± 0.02	0.58 ± 0.01
	PACAP + HS	0.59 ± 0.01 #	0.56 ± 0.01 *	0.57 ± 0.01 *	0.59 ± 0.01 +&
SS_1/2_ [Pa]	Control	1.51 ± 0.20	1.50 ± 0.12	1.58 ± 0.18	1.54 ± 0.22
	HS	1.47 ± 0.20	1.56 ± 0.14	1.67 ± 0.21	1.54 ± 0.15
	PACAP	1.47 ± 0.19	1.45 ± 0.13	1.63 ± 0.32	1.48 ± 0.23
	PACAP + HS	1.61 ± 0.13	1.61 ± 0.18 &	1.46 ± 0.17 *+	1.59 ± 0.16
M 5s	Control	2.64 ± 1.04	3.87 ± 1.46 *	3.14 ± 0.82	2.78 ± 1.01
	HS	3 ± 1.35	4.44 ± 1.21 *	3.90 ± 1.55	3.88 ± 2.16
	PACAP	3.62 ± 1.34 #	4.97 ± 1.83 *	4.51 ± 1.37 *#	5.15 ± 1.67 *#
	PACAP + HS	3.09 ± 1.05	4.56 ± 1.63 *	4.67 ± 1.31 *#	3.68 ± 0.98 #&
M 10s	Control	6.39 ± 3.81	8.31 ± 2.42	7.84 ± 2.04	6.87 ± 1.55
	HS	7.45 ± 2.70	10.38 ± 3.81 *#	9.17 ± 4.12	9.75 ± 2.41 *#
	PACAP	8.89 ± 2.59 #	11.24 ± 3.58 *#	9.10 ± 2.60	9.22 ± 2.53 #
	PACAP + HS	9.32 ± 2.41 #	9.53 ± 3.02	9.16 ± 2.04 #	8.60 ± 1.47 #
M1 5s	Control	2.41 ± 1.41	3.01 ± 1.51	2.93 ± 0.95	2.14 ± 1.09
	HS	2.48 ± 1.02	4.08 ± 1.56 *	2.90 ± 0.95	3.03 ± 1.63
	PACAP	3.43 ± 1.30 #+	4.08 ± 1.37 *#	3.32 ± 1.29	4.05 ± 1.16 *#+
	PACAP + HS	3.20 ± 1.08 +	3.78 ± 1.53	3.69 ± 1.18 #+	3.29 ± 0.98 #&
M1 10s	Control	6.06 ± 3.62	7.83 ± 3.64	7.22 ± 2.21	5.89 ± 1.37
	HS	6.90 ± 3.11	9.82 ± 4.43 *	8.92 ± 4.13	8.23 ± 2.82 #
	PACAP	9.30 ± 2.66 #+	10.82 ± 3.95 #	8.47 ± 2.86	9.67 ± 2.61 #
	PACAP + HS	9.02 ± 20 #+	9.15 ± 3.36	9.77 ± 2.89 #	8.39 ± 2.81 #

Means ± S.D., * *p* < 0.05 vs. base, # *p* < 0.05 vs. Control, + *p* < 0.05 vs. HS, and & *p* < 0.05 vs. PACAP.

**Table 3 ijms-24-16695-t003:** Elastin content in freshly excised and tissue-cultured vessels.

Variable	Group	Control Side	Anastomosis
**Freshly excised arteries**			
Integrated green light density [number of pixels] median (Q1–Q3)	Control	13,501.88 (13,205.36–197,852.10)	14,731.83 (13,597.90–224,739.80)
	HS	81,904.64 (13,576.89–168,056.55)	1587.68 (1117.11–14,289.80)
	PACAP	15,613.63 (14,051.40–149,313.40)	13,520.22 (13,218.87–14,900.47)
	PACAP + HS	13,912.72 (13,437.50–106,210.71)	13,516.40 (13,218.87–14,149.79)
Number of elastic membranes [number/field] means ± S.D.	Control	3.50 ± 0.58	4 ± 1.41
	HS	4.20 ± 0.44	3 ± 1
	PACAP	4.86 ± 1.21	5.40 ± 1.34
	PACAP + HS	5.00 ± 1.58	3.33 ± 1.15
Western blot integrated pixel density [number of pixels] means ± S.D.	Control	1 ± 0	0.6 ± 0.0
	HS	1 ± 0	1.05 ± 0.07
	PACAP	1 ± 0	1.6 ± 0.0
	PACAP + HS	1 ± 0	2.6 ± 0.0
**Tissue cultured arteries**			
Integrated green light density [number of pixels] median (Q1–Q3)	Control	13,850.36 (13,566.50–48,305.46)	13,871.32 (13,486.03–166,405.70)
	PACAP	13,912.72 (13,396.37–14,600.46)	14,511.32 (14,076.57–185,590.30)
Number of elastic membranes [number/field] means ± S.D.	Control	3.50 ± 0.55	3.75 ± 0.96
	PACAP	4.80 ± 1.30	4.67 ± 0.58

**Table 4 ijms-24-16695-t004:** Collagen type I content in freshly excised and tissue-cultured vessels.

Variable	Group	Control Side	Anastomosis
**Freshly excised arteries**			
Red light intensity [number of pixels] median (Q1–Q3)	Control	13,825.44 (9343.24–20,063.81)	9291.28 (7364.89–14,288.00)
	HS	9322.30 (7509.55–22,460.15)	7120.21 (5930.57–7931.24)
	PACAP	12,484.11 (8666.51–13,373.92)	10,621.18 (8903.27–11,105.26)
	PACAP + HS	13,198.04 (11,975.60–13,240.30)	9597.01 (7794.21–10,650.40)
Green light intensity [number of pixels] median (Q1–Q3)	Control	1473.62 (1270.92–8565.78)	3799.73 (1564.56–6101.55)
	HS	4389.07 (4042.19–9744.79)	3677.36 (2871.04–4479.74)
	PACAP	2787.34 (2182.58–4455.78)	3223.48 (2049.68–3928.61)
	PACAP + HS	4632.54 (1583.69–4829.60)	1979.14 (1846.70–2104.08)
Western blot integrated pixel density [number/field] means ± S.D.	Control	1 ± 0	0.4 ± 0
	HS	1 ± 0	1 ± 0
	PACAP	1 ± 0	0.2 ± 0
	PACAP + HS	1 ± 0	0.5 ± 0
**Tissue cultured arteries**			
Red light intensity [number of pixels] median (Q1–Q3)	Control	41,106.71 (22,098.34–43,538.714)	21,036.31 (8323.01–21,060.99)
	PACAP	65,156.94 (6218.90–77,448.39)	12,360.17 (12,281.60–12,473.94)
Green light intensity [number of pixels] median (Q1-Q3)	Control	4767.25 (4491.69–5700.38)	5973.45 (2862.03–6410.84)
	PACAP	3199.41 (1657.89–8563.06)	2092.74 (1686.29–5685.69)

## Data Availability

The data presented in this study are available on request from the corresponding author. The data are not publicly available due to ethical permission.

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
