# Peer review of "Impact Assessment of Pituitary Adenylate Cyclase Activating Polypeptide (PACAP) and Hemostatic Sponge on Vascular Anastomosis Regeneration in Rats"

_ijms, 2023, doi:10.3390/ijms242316695_

Round 1
Reviewer 1 Report
Comments and Suggestions for Authors
The current study demonstrated a comparison of the effects of local administration of PACAP and/or bioplast on the regeneration of end-to-end microvascular anastomoses in rats. The working hypothesis was that local administration of PACAP may facilitate the regeneration of end-to-end microvascular anastomoses, and the effect can be enhanced when PACAP and bioplast application are combined. The primary objectives of researchers were to explore alterations in hematological, hemorheological, histological, and biomechanical features of the regenerated end-to-end anastomosis in a longitudinal study involving rats.
The findings of the study showed a decrease in arterial wall thickness when bioplastic was present, which was compensated by the application of PACAP. Comparable outcomes were noted in the tunica media and adventitia in vivo. Administration of PACAP increased these parameters in vitro. The use of PACAP increased the expression of elastin, while the bioplast usage led to its reduction. However, the expression of type I collagen remained largely unchanged. Their combined use was beneficial for vascular regeneration.
General concept comments
The authors focused on comparing the effects of local PACAP and/or bioplast administration on the regeneration of end-to-end microvascular anastomoses in rats, but the title of the article does not reflect this. The hypothesis is interesting and innovative, but the purpose of the work is incorrectly formulated. The aim of the work is too general and presents the methods the authors will use rather than the parameters they intend to check the testability of the hypothesis.
Methodologically, the authors carry out numerous in vivo and in vitro tests, as well as laboratory tests, immunohistochemistry, measurements of the tensile strength of anastomoses, and others. The experimental design is appropriate to test the hypothesis.
However, the manuscript is not written in good English and requires more than minor English editing before publication. Moreover, the authors use incorrect nomenclature for the reference substance used. The name "Bioplast" refers to a hemostatic sponge, specifically, Spongostan was used in the study (without explaining why this product and not Medisponge, Vliwasorb, SUPRASORB P or other products from other companies). As a side note, please check whether the manufacturer of Spongostan has been provided correctly.)
Specific comments
I found the name Bioplast in works from 1977 and 1980. It seems that the local name of the hemostatic sponge and perhaps the term Spongostan should not be used in a scientific article. https://www.sciencedirect.com/science/article/abs/pii/S030097857880074X
https://www.sciencedirect.com/science/article/abs/pii/0142961280900551
The name Bioplast is written sometimes in lowercase and sometimes in capital letters - you should choose one form and use it consistently.
There is no information about the source from which PACAP was obtained.
Lines 20-25: The sentences need to be rephrased.
Not only the Abstract requires language correction. In the description of the Results, there are many phrases from everyday language, for example: “The respiration of the animals during anesthesia was similar in all groups”; “Red blood cell deformability showed a slight worsening”
Line 27: “The effect of PACAP was also investigated by in vitro tissue culture” - The method was indicated, not what parameters were tested in tissue culture in vitro.
Line 337: Not the effect of PACAP but the comparison of the effect of PACAP and/or Spongostan on the regeneration of vascular anastomoses was investigated.
Line 654: “10 nM” - S.I. units should be used. Other cases of incorrectly used units should be checked and corrected.
Too many Reference items and many cited items lack references to the text. For example, in item 16 - there is no reference to a hemostatic sponge in the entire article. 52 references were published more than 5 years ago, which constitutes 60% of all references, and 37 more than 10 years ago - 36% of all references
In Conclusions that highlight the most important achievements of the work, information, and results obtained that are not significant should not be included in the last sentence (lines 733-735).
The figures are not presented correctly: In Figure 1, the Y-axis is not properly described. - A) "Newton (N)"; B) and C) working shortcuts?
Figure 4 is not on the same page as its caption.
The abbreviation BFU is only explained on line 606, but without a description of what it is, and first appears on line 127.
The abbreviation H&E is only explained on line 664 (page 18) and first appears in the caption of Figure 2 on page 6.
The terms in vitro and in vivo are written inconsistently (sometimes in italics and sometimes not).
The authors should add a list of abbreviations that might improve it for readers and not propose abbreviations for names that are presented only once.

The manuscript requires extensive editing of the English language.
Author Response
Dear Reviewer,
thank you very much for your time to prepare the review, with helpful and valuable comments. In the revised version the corrections and additions have been made, according to the comments. Below, please find the point-by-point responses.
English language revision has been made. We have corrected the relevant sentences in the Abstract, too. We modified the sentence on line 337.
The hypothesis and study aim have been clarified, hopefully better.
Concerning the word 'bioplast', we used an old-school term. We have exchanged it to the 'hemostatic sponge' (HS as abbreviation) in the entire text, including the figures and tables.
We added the information about the source of the PACAP in the Materials and methods session 4.2.
Thank you for your comment on the SI units. We corrected the ‘nM’ to ‘nmol’.
Related to vascular anastomosis and tissue regeneration we could exchange several references to more recent ones.
The figure description has been modified, and the data shown in Figures B and C have been explained in detail (Materials and methods session 4.5).
Figure 4 and its caption were placed on the same page in the revised version.
Blood flux unit abbreviation as BFU has been added to Materials and methods session 4.4.
We corrected the in vitro and in vivo terms to italics everywhere.
We added an abbreviation list. The abbreviation H&E has been added to the abbreviation list.
We hope that the responses could be acceptable, and the revised version could be improved. We express our thanks again for the Reviewer’s comments, which were valuable and helpful.
Sincerely yours,
Norbert Nemeth
Reviewer 2 Report
Comments and Suggestions for Authors
There are a considerable number of issues in the manuscript by Fazekas et al.
Introduction
-Reference 10 does not describe “anastomoses of the bowel” (line 61). Please check all references for accuracy.
Results
-PACAP possesses anti-inflammatory activities. How to explain the result described in lines 142-145 (Table 1)?
-Orcein staining of elastin shows purple. Why is “green” light density shown in Table 3?
-In Tables 3 and 4, some SDs of staining intensity are similar to or greater than means, indicating non-normal distributions of the data that should be presented as median (the interquartile range) and analyzed with a different statistic method.
-Effects of PACAP on vessel wall thickness in teared anastomosis are discordant in in vitro and in vivo studies as shown in Figs 2 and 3. Please explain.
-In Table 3, in vivo results show inhibitory effects of PACAP on elastin expression in anastomosis while in vitro data show that PACAP slightly increases elastin expression. Please explain.
-In Tables 3 and 4, regarding Western blotting data, please present the density of the target gene band normalized against that of the house keeping gene band and provide the relevant statistical analysis result.
-Elastin immunostaining (Fig 5) in control anastomosis and anastomosis treated with PACAP demonstrates results contradicting those shown in Table 3.
-Please change “Table 3” that shows collagen data to “Table 4”.
-In Fig S3, all protein bands of the house keeping gene in the intact or anastomosed vessel in 4 groups have a density value of 1.0, but the band density from band to band looks substantially different. The expression level of the target gene normalized against the house keeping gene should be statistically analyzed. Also, I suggest placing this figure and Western blotting results in the main text.
Materials and Methods
-For PACAP+BP animals, I assume the BP will prevent PACAP from reaching the anastomosis site and reduce the drug effect.
-What is the size of the hole cut in the cannula for local drug administration?
-Will the cannula and bioplast placed around the anastomosis compress the femoral artery to promote flow disturbance?
-Please change “Pre-operatively, on days 7, 14, and 21 postoperatively, a lateral tail vein was cannulated” (lines 572-573) to “Pre-operatively, and on days 7, 14, and 21 postoperatively, a lateral tail vein was cannulated”.
-Why was intravenous ketamine/xylazine given after vessel removal not before? (lines 581-582).
-Regarding “a portion of the severed blood vessels was cultured in HG-DMEM medium” (line 648), please specify the portion of the vessel and its length (proximal or distal to the anastomosis, or the anastomosis part)?
-According to Fig 9, the first third of the curve (red) is above 0.0098 N but below 0.2 N, which was not included in the slope calculation. Lines 638-643 should be modified to reflect this.
- Regarding “some samples were treated with PACAP” (line 653), please specify the samples. Are the samples derived from animals already treated with PACAP or not?
-Were arteries (lines 656-660) cultured in the same manner as described in lines 647-655?
-A total of 32 rats had anastomosis and were divided into 4 groups. What are the two remaining rats (“From the two remaining rats”, line 658)?
Conclusions
-Inconsistent results are very concerning. More animals may be required for the study.
English editing is required for clarity or the correction of mistakes, e.g., what is the “experimental method” (line 60)? “involves” (line 71) should be “involve”, etc.
Comments on the Quality of English Language
Minor English editing is required.
Author Response
Dear Reviewer,
thank you very much for your time to prepare the review, with helpful and valuable comments. In the revised version the corrections (including grammar revision) and additions have been made, according to the comments. Below, please find the point-by-point responses.
In line 61 we corrected the sentence related to anastomoses.
Related to the comment on PACAP anti-inflammatory activities and the findings we would say that the controversial results between the in vivo and in vitro experiments due to the lack of the systemic PACAP degrading enzymes may alter the local effects of the neuropeptide. Although it further proved the unquestionable effects of PACAP on vessel regeneration. It has been included Discussion.
The orcein staining description has been clarified in the Materials and Methods session 4.7. We added the following sentences: In orcein staining the normal color inverted to green and black to increase the contrast of the pixels. Green pixel number was determined by ImageJ software. Equal area of vessel wall of 20x objective magnification was used to determine orcein (green) positivity and were shown in graphs.
Although data were uniformly presented, when data showed greater SDs, indicating non-normal distributions, the relevant statistical tests were used as explained in session 4.10.
We used actin as a housekeeping protein and the results of elastin and collagen type I expression were normalized on the expression of actin separately to control, HS (hemostatic sponge instead of the term ‘bioplast’) to HS, PACAP to PACAP and PACAP+HS to PACAP+HS. Statistical analysis between the experimental groups were determined after the normality test.
Related to the comment on results showed on Figure 5 and Table 3, we would like to respond that the orcein staining is specific for elastic fibers which shows properly the visible fibers or lamella system. With immunhistochemistry even the pre-elastin fibers can be labelled, furthermore the RBC shows very strong false positive signals. Therefore the stronger immunpositivity in PACAP treated anastomosis could be responsible for the results of intracellular elastin positivity, while in Table 3 only extracellular fully formed fibers were evaluated.
Table 3 has been changed to Table 4. Thank you!
Thank you for your suggestions on Fig. S3, we added the information. The main text is very long with numerous figures and tables. This was the reason why we placed the Western blotting results into the Supplementary file.
Concerning the effect of hemostatic sponge, thank you for your suggestion. However, our results also suggest the hemostatic sponge and PACAP provide a stronger and favorable result which suggest the PACAP easily pass via the hemostatic sponge. On the other hand, this topical agent can work as an absorptive material, which can prolong the effect of PACAP, supposedly preventing the neuropeptide from the degradation.
The size data of the cut hole in the cannula have been added in session 4.2.
Thank you for your question on possible compressing effect of the hemostatic agent placed around the vessel. Before the epigastric branches the femoral blood vessels are located in an imaginary anatomical valley between the muscles (quadriceps femoris and gracilis) and the cannula was passed perpendicularly over these muscles, so that it could not put any compression on the vessels. At the end of the study, when the blood vessels were dissected and observed, no constriction was observed and the flow was adequate. Observing the legs of the animals, atrophy was seen only in the thrombosed cases.
The sentence in lines 572-573 have been modified. Thank you!
The ketamine/xylazine administration after vessel removal has been clarified: We gave it only at the very end to prevent the high concentrated solutions affecting on the vessels.
Thank you for the question regarding the vessel portions and lengths. The artery pieces were similar in size to the arteries torn into two pieces (approximately 1 cm) for the experiments equally proximal and distal part of vessels were used. During the in vitro experiments we continued the same treatments on certain vessels as in vivo, except for benchmark group which was not torn and treated in vivo. We added this information in the revised text.
We have modified the Fig 9 legend and the related methodology part. Due to the irregularity of the initial curve part, these data (below 0.0098 N) were removed from the analysis range.
The sample description for PACAP treatments has been clarified. Thank you!
The arteries were cultured in the same manner as described in lines 647-655.
Thank you for your comment on the animal number. The additional two animals provided the benchmark arteries, which were not treated in vivo. The femoral arteries from these two animals (2/each) were cut into proximal and distal sections according to the anastomoses, and then equally divided (control and PACAP treated).
We agree with the Reviewer, that more animals may be required to clarify the findings. As we wrote in the Discussion, one of the limitations of the study is the relative low case number. Based on the findings we plan to conduct further studies, focusing on specified details.
We hope that the responses could be acceptable, and the revised version could be improved. We express our thanks again for the Reviewer’s comments, which were valuable and helpful.
Sincerely yours,
Norbert Nemeth
Round 2
Reviewer 1 Report
Comments and Suggestions for Authors
Regarding the title, please consider a phrasing that carries a stronger impact than the sentence proposed by the authors..:
“Comparison of the effect of the application of pituitary adenylate cyclase activating polypeptide (PACAP) and hemostatic sponge on the regeneration of vascular anastomosis in rats” or
“Impact Assessment of Pituitary Adenylate Cyclase Activating Polypeptide (PACAP) and Hemostatic Sponge on Vascular Anastomosis Regeneration in Rats”
Regarding the SI units: it appears that the authors have primarily addressed the highlighted example and have not fully implemented the recommended changes.: „Line 654: “10 nM” - S.I. units should be used. Other cases of incorrectly used units should be checked and corrected.” Please take a moment to review your manuscript carefully. The unit of concentration corresponding to M is mol/L, not simply mol ( and nmol/L or mmol/L, respectively).
Comments on the Quality of English LanguageDespite the initial suggestion: “However the manuscript is not written in good English and requires more than minor English editing before publication” and the subsequent response: “English language revision has been made.”, it appears that the authors have primarily addressed only the specific instances of awkward or overly informal language highlighted by the reviewer. The manuscript, as it stands, still requires improvement in its English language usage. If the authors find this task challenging, it would be beneficial to utilize the publisher-recommended website for assistance. https://www.mdpi.com/authors/english
Author Response
Dear Reviewer,
Thank you very much for your valuable comments on the revised version of our manuscript. We are very glad that based on your suggestions we could improve it. We have just completed the English revision. The SI units have been corrected in the entire manuscript. Thank you very much for the suggestions on phrasing the title. We modified it, accordingly.
We hope that the corrections can be acceptable.
Sincerely yours,
Norbert Nemeth
Reviewer 2 Report
Comments and Suggestions for Authors
I appreciate the efforts made by the authors in revising the manuscript. However, some of my concerns are not appropriately addressed.
Results
1) Please present all non-parametric data as median (the interquartile range).
2) In Fig S3, for freshly excised arteries the density of actin between the intact and anastomosed vessel within the same group looks substantially different. Please provide new blots showing a similar actin density between the intact and anastomosed vessel in the same group to replace the current ones.
Discussion
1) Please change “The controversial results between the in vivo and in vitro experiments due to the lack of the systemic PACAP degrading enzymes may alter the local effects of the neuropeptide. Although it further proved the unquestionable effects of PACAP on vessel regeneration” to “Although this study proved the unquestionable effects of PACAP on vessel regeneration, the controversial results between the in vivo and in vitro experiments may be due to the presence and absence of PACAP degrading enzymes in vivo and in vitro respectively, leading to different local effects of the neuropeptide”.
Materials and Methods
1) Are in vitro assays done in a blind manner? Please clarify.
2) Please change “Due to the irregularity of the initial curve part was removed (below 0.0098 N)” (line 642) to “The first one-third of the curve (red) was not included in the slope calculation due to its irregularity”.
3) Please delete “first” in line 645.
4) Please remove “separately Control to Control, HS to HS PACAP to PACAP and PACAP+HS to PACAP+HS” (lines 729-730).
Conclusions
1) Please remove “and their combined use sometimes enhanced and sometimes reversed their effects” (lines 742-743).
Comments on the Quality of English Language
English editing is required.
Author Response
Dear Reviewer,
Thank you very much for your valuable comments on the revised version of our manuscript.
Non-parametric data have been presented, accordingly.
For Fig. S3 we found a better representative picture, thank you very much.
The highlighted points for clarification, phrasings and other corrections have been all made.
We hope that the correction can be acceptable.
Sincerely yours,
Norbert Nemeth
Round 3
Reviewer 2 Report
Comments and Suggestions for Authors
The manuscript is significantly improved. “BP” for bioplast still appears in the article (Table 1 and lines 366, 568, 757, and 759). Please correct the mistakes.
Author Response
Dear Reviewer, thank you very much! We corrected these mistakes.
Sincerely yours, Norbert Nemeth